# Major depressive disorder and bistability in an HPA-CNS toggle switch

**Ben Ron Mizrachi, Avichai Tendler, Omer Karin, Tomer Milo, Dafna Haran, Avi Mayo, Uri Alon** *

Dept. Molecular Cell biology, Weizmann Institute of Science, Rehovot, Israel

* urialonw@gmail.com

## Abstract

Major depressive disorder (MDD) is the most common psychiatric disorder. It has a complex and heterogeneous etiology. Most treatments take weeks to show effects and work well only for a fraction of the patients. Thus, new concepts are needed to understand MDD and its dynamics. One of the strong correlates of MDD is increased activity and dysregulation of the hypothalamic-pituitary-adrenal (HPA) axis which produces the stress hormone cortisol. Existing mathematical models of the HPA axis describe its operation on the scale of hours, and thus are unable to explore the dynamic on the scale of weeks that characterizes many aspects of MDD. Here, we propose a mathematical model of MDD on the scale of weeks, a timescale provided by the growth of the HPA hormone glands under control of HPA hormones. We add to this the mutual inhibition of the HPA axis and the hippocampus and other regions of the central nervous system (CNS) that forms a toggle switch. The model shows bistability between euthymic and depressed states, with a slow timescale of weeks in its dynamics. It explains why prolonged but not acute stress can trigger a self-sustaining depressive episode that persists even after the stress is removed. The model explains the weeks timescale for drugs to take effect, as well as the dysregulation of the HPA axis in MDD, based on gland mass changes. This understanding of MDD dynamics may help to guide strategies for treatment.

**Data Availability Statement:** Code files, animations, generated by Mathematica, of the phase portraits under changing u, a or b parameters can be found here: https://github.com/

## Author summary

Major depressive disorder (MDD) is a prevalent psychiatric condition whose mechanisms are not fully understood. Existing treatments often take weeks to be effective and only benefit a portion of patients, necessitating a fresh perspective on understanding MDD. Research has shown a strong association between MDD and the dysregulation of the human stress hormone pathway. However, previous mathematical models of the stress hormone pathway have only explored its dynamics over a few hours or days, not capturing the weeks-long timescale relevant to MDD. To address this, we present a new mathematical model that incorporates the growth of the stress hormone glands, and their mutually inhibitory interactions with the brain. This model reveals how the brain and the stress axis toggle between normal and depressed states, with slow dynamics over weeks. It

benron20/MDD-and-bistability-in-the-HPA-stress-axis.

**Funding:** This project was funded by the European Research Council (ERC) under the European Union's Horizon 2020 research and innovation program (grant agreement No 856487). The funders had no role in study design, data collection and analysis, decision to publish, or preparation of the manuscript.

**Competing interests:** The authors have declared that no competing interests exist.

also explains why prolonged stress—but not brief stress—can trigger a sustained depressive episode. The model further sheds light on why antidepressant drugs take weeks to take effect. By comprehending the dynamics of depression, this model may provide valuable insights for guiding treatment strategies in the future.

## Introduction

Major depressive disorder (MDD) is the most common psychiatric syndrome, affecting about 20% of the world population at some point in life [1]. There were 21 million recorded cases of depression in adults in 2020 in the USA, about 8% of the adult population [2]. MDD in adolescents and children has risen sharply in the past decade in many countries [3]. People with MDD suffer from depressed mood or sadness, anhedonia, and may suffer from sleeping and eating disorders. The etiology of MDD is complex and heterogeneous, and multiple genetic and environmental factors contribute to its incidence [4].

Treatment for MDD includes medication, psychotherapy, physical exercise, and electroconvulsive therapy (ECT). Treatment is usually effective for only a fraction of the patients [5,6] and often takes several weeks to show effects [7]. In the past five decades few novel therapeutics for MDD have emerged that go beyond the monoamine theory of depression first developed in the 1960s [8].

Among the endocrine, inflammatory, metabolic and neuronal factors that have been explored, the stress hormone cortisol is one of the most significant correlates of MDD [9]. Stress is associated with MDD, and cortisol is higher than controls in most studies of people with MDD [10]. High cortisol is not only a readout of MDD—it can also be causal for depression. This is seen in Cushing's syndrome in which a tumor induces high levels of cortisol, resulting in depression in over 50% of patients [11–13]. Similarly, prolonged high-dose glucocorticoid treatment, such as the cortisol analogues dexamethasone (DEX) or prednisone taken for a few weeks or longer, is associated with depression in a dose-dependent manner [14].

Furthermore, the endocrine system that secretes cortisol, the hypothalamic-pituitary-adrenal (HPA) axis, is dysregulated in many patients with MDD [15]. Enlarged adrenal glands in suicide victims and stressed animals were one of the clues that led to characterization of the HPA axis [16]. HPA dysregulation in patients with MDD is evident in enlarged adrenal glands [17,18], blunted adrenocorticotropic hormone (ACTH) response in corticotropin-releasing-hormone (CRH) tests [19] and elevated ACTH response in DEX-CRH tests [20,21].

MDD also involves dysfunctions of the central nervous system (CNS) such as impaired memory and hippocampal atrophy [22–24]. The CNS and HPA have mutual interactions involved in MDD. The hippocampus and the prefrontal cortex have inhibitory effects on HPA activation [25–27]. Conversely, high levels of cortisol can have inhibitory effects on these CNS regions. Cortisol is sensed by glucocorticoid receptors (GR) in the hippocampus and prefrontal cortex, causing reduced neurogenesis, synaptic arbor reduction, neuronal death [28,29], and a long-term decrease in CNS volume [30]. This interplay between the CNS and HPA is believed to be mechanistically important for MDD [26].

There are several open questions concerning the origin and dynamics of MDD. How does prolonged but not acute stress trigger depression? Once triggered, how does depression persist even after the stressor is gone? What is the origin of the timescale of weeks or longer of each depressive episode as well as the timescale for drugs to take effect? What is the origin of the difference between individuals susceptible to MDD and those not susceptible, in which a given

stress triggers depression in the former and not the latter? Understanding these questions can help to guide MDD research and treatment.

One way to explore these open questions is to use mathematical modeling [31,32]. Some of the previous mathematical models of MDD did not include specific physiological mechanisms. They show recurrence of depressive episodes [33]. More mechanistic models based on the HPA axis describe the dynamics of hormones and their receptors on the scale of hours [34–43]. Some show the interplay between the CNS and HPA [39], and present bistability dynamics as an explanation of stress vulnerability and MDD [36,41,42,44]. These HPA models do not discuss the timescale of weeks which is seen in many aspects of MDD.

To address the dynamics of MDD on the scale of weeks, we consider the interplay between the HPA and the CNS within a minimal mathematical model. The model supplies the relevant timescale of weeks by including the slow growth of HPA glands under control of HPA hormones, an approach recently developed in our group [45–47]. We add to this the mutual inhibition by the CNS to obtain a new model which describes the mechanism of depression as a bistable system with a slow dynamical timescale of weeks. The model can explain why prolonged but not acute stress triggers depression, why treatment can take weeks, and why the HPA axis is dysregulated in MDD. It also reveals a hysteresis property that makes recovery from depression more difficult than entry into it.

## Results

### Toggle switch model of the HPA—CNS system

We present a minimal mathematical model of the interaction between the HPA axis and the CNS processes that inhibit it (Fig 1A). The model has two variables: the functional mass of the cortisol-secreting cells in the adrenal cortex, $A$, and the inhibitory activity of the CNS, $h$. The inhibitory activity $h$ is inspired by the hippocampus, which inhibits the hypothalamic inputs to the HPA axis [26]. Our approach is agnostic, however, to the exact nature of the CNS variable $h$, which could comprise in addition to the hippocampus, also other regions including the prefrontal cortex [27,48].

The main idea is that cortisol produced by the HPA axis acts through the glucocorticoid receptors (GR) [30,49] to inhibit the inhibitory CNS activity $h$ [26]. This activity, in turn, inhibits the HPA axis. The mutual inhibition creates a toggle-switch circuit: $A$ and $h$ inhibit each other (Fig 1B). Since the adrenal cortex functional mass $A$ changes over weeks, a slow timescale is introduced to the dynamics of the system.

The model can be compactly understood by the nullcline method, an approach which separates the toggle switch into two arms [50,51]. We first treat each arm independently, and then combine them to find the fixed points of the system, given by the crossing points of the nullclines.

The conclusions are robust in the sense that they depend only on the qualitative shapes of the nullclines and not on their precise mathematical forms. This is useful for MDD in which many of the physiological mechanisms are not yet fully characterized. In addition to the qualitative approach, we describe in Methods a specific mathematical model based on a model for the HPA axis [45] together with equations for the CNS inhibitory activity $h$. The Methods section details the equations and parameters used to simulate the dynamics.

The first nullcline describes the inhibition of the HPA axis by the CNS. It is defined by the adrenal cortex mass steady state, $\frac{dA}{dt} = 0$, as a function of the inhibitory activity $h$. $h$ inhibits the HPA axis, since a large $h$ reduces CRH [26] the inducer of ACTH which is the growth factor for the adrenal cortex $A$. If one clamps $h$ at a given value, $A$ changes its functional mass over the timescale of weeks to reach a steady state $A_{st}$. The higher $h$, the lower the HPA activity and

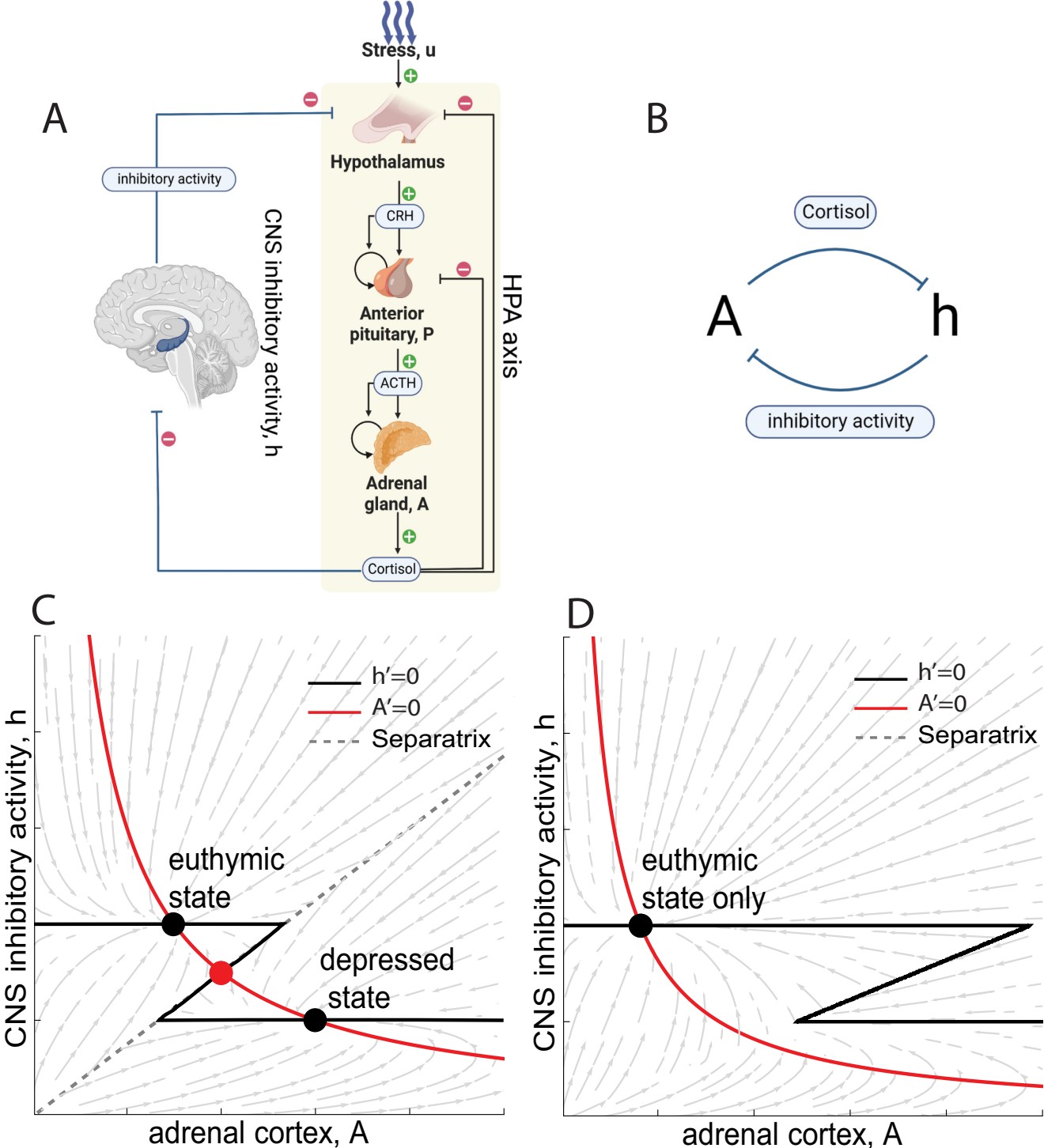

**Fig 1. Mathematical model for the interactions between the HPA axis and its inhibitory CNS regions.** A) Schematic of HPA and CNS interactions in the model (icons taken from Biorender [52] according to their Academic License Terms [53]). B) The model can be summarized as a toggle switch between CNS inhibitory activity $h$ and the adrenal mass, $A$. CNS inhibits the hypothalamic input and thus leads to a decrease in $A$. $A$ releases cortisol, which inhibits CNS through the glucocorticoid receptor GR. C) Nullclines show three fixed points in susceptible individuals. Euthymic and depressed states are marked in black dots. D) In non-susceptible individuals there is only one euthymic fixed point. C and D used identical parameters except stress input $u = 0.75$ and $u = 0.4$ respectively. An analogous shift between 3 and 1 fixed points can occur when changing the CNS system parameters to represent genetic variants. Figure created with Biorender.

hence the smaller $A_{st}$. This provides a decreasing curve plotted in the phase plane whose axes are $A$ and $h$ (Fig 1C, red line).

In the model described in Methods, solving Eq (18) gives a specific form for this nullcline, $A \propto \frac{u}{h}$ so that $A$ drops with $h$. Note for future reference that adrenal mass $A$ rises with the stress input $u$.

The second arm of the toggle switch, in which $A$ inhibits $h$, is given by steady state CNS activity $h_{st}$ as a function of $A$. The steady state is the value of $h$ at which $\frac{dh}{dt} = 0$. Due to the effects of cortisol, $h$ declines with $A$. Since cortisol inhibits the activity of relevant brain regions such as the hippocampus and the prefrontal cortex through the GR receptor, which has a cooperative (step like) activity dependence on cortisol, we assume that $h$ goes from a high level $h_{high}$ to a low level $h_{low}$ in a step-like manner (Fig 1C, black line) when cortisol produced by $A$ crosses the dissociation constant $k_d$ of the GR receptor.

The HPA-CNS model gives the steady state solution $h_{st} = \frac{1}{GR(x_3)}$, which is a step-like function due to the high cooperativity of the GR receptor with a Hill coefficient of about 3.

The qualitative shapes of the nullclines are more general than the specific model equations and suffice for many of the conclusions below.

## Model shows bistability between euthymic and depressed states

There are two main scenarios for the model nullclines, which correspond to individuals susceptible and resilient to MDD. At the points where the nullclines intersect, the system is at steady state, because both $\frac{dA}{dt} = 0$ and $\frac{dh}{dt} = 0$.

In the first scenario, the two nullclines intersect at three fixed points (Fig 1C). One fixed point has low adrenal cortex mass, $A$, and high CNS inhibition, $h$, and can be considered as the euthymic or healthy state. A second fixed point has high adrenal mass $A$ and low CNS inhibition $h$, and represents the depressed state, with high cortisol and an enlarged adrenal cortex. In between, there is an unstable fixed point. This system is bistable: depending on initial conditions, it reaches either the healthy or the MDD state. There is a basin of attraction for each state. The boundary between these basins is a curve called the separatrix which goes through the unstable fixed point.

In the second scenario, the parameters of the nullclines are such that there is only a single intersection point (Fig 1D). This is the healthy state with low A—the MDD state with high $A$ no longer exists. Such individuals are resilient with respect to major depressive disorder. All initial conditions eventually flow to the healthy fixed point.

Susceptibility to MDD is partly due to genetics [11,14]. To conceptualize this susceptibility using the model we note that most MDD-related mutations are not in HPA-related genes (except for genes related to GR), and thus we may assume that the $A$ nullcline is similar between individuals. On the other hand, the $h$ nullcline may depend on many genetic variants that affect the CNS. Multi-gene variations might push this nullcline up or down, and contribute to the hereditary component of susceptibility or resilience to MDD [54]. We analyzed which parameters changes most sensitively change the number of fixed points (Table 1).

## Prolonged but not acute stress can trigger self-sustained depression

The model can help understand the differential effects of prolonged and acute stress. Stress can be modeled by an increase in hypothalamic input $u$. This shifts the $A$ nullcline up and to the right—because at a given level of CNS inhibition, high stress leads to an enlarged adrenal relative to the low stress condition. It also shifts the $h$ nullcline to the left—because higher stress is more likely to activate GR receptors in the CNS. In the bistable case, a large enough

**Table 1. HPA-CNS model nominal parameter values and sensitivities.**

| Parameter | meaning | Nominal value | Fold-change for bifurcation to 1 fixed point |
|---|---|---|---|
| a | CNS inhibition of HPA at low cortisol | 1 | 0.5,1.5 |
| b | CNS inhibition of HPA at high cortisol | 1 | 0.5 |
| u | stress input | 1 | 0.75,1.5 |
| T | GR activation threshold in the CNS | 1.5 | 0.66, 1.33 |
| $a_1$ | CRH secretion parameter | 0.17 CRH*cortisol*h/min | 0.75, 1.5 |
| $b_1$ | CRH removal rate | 0.17/min [39] | 0.66, 1.33 |
| $a_2$ | ACTH secretion parameter | 0.035/min/(CRH*P) | 0.75, 1.5 |
| $b_2$ | ACTH removal rate | 0.035/min [39] | 0.66, 1.33 |
| $a_3$ | cortisol secretion parameter | 0.0086/min/(ACTH*A) | 0.7, 1.45 |
| $b_3$ | cortisol removal rate | 0.0086/min [39] | 0.68, 1.42 |
| $a_P$ | pituitary cell-mass production per unit CRH | 0.049/day/CRH | 0.85, 1.2 |
| $b_P$ | pituitary cell-mass removal rate | 0.049/day [45] | 0.83, 1.17 |
| $a_A$ | adrenal cell-mass production per unit ACTH | 0.099/day/ACTH | 0.85, 1.2 |
| $b_A$ | adrenal cell-mass removal rate | 0.099/day [45] | 0.83, 1.17 |
| $a_h$ | CNS production parameter | 0.047 h/day | 0.65, 1.35 |
| $b_h$ | CNS removal rate | 0.047/day | 0.74, 1.54 |

shift can abolish the healthy fixed point, leaving MDD as the only possible steady-state (Fig 2A1).

If stress lasts less than a few weeks, *A* starts growing but does not reach the separatrix. After stress is over, the nullcline shifts back to produce a healthy fixed point again, and *A* returns to its healthy size within weeks. No long-term depression develops (Fig 2A2).

If stress however lasts longer than a few weeks, *A* can grow so much that it crosses the original (pre-stress) separatrix. Now even if stress is over, and the healthy fixed point is reinstated, A continues to flow to the MDD fixed point because it lies within its basin of attraction. Once it reaches the MDD fixed point, the system stays there even if stress returns to its previous low value. Therefore, the MDD fixed point is self-sustaining, with low inhibition that fuels high cortisol, which in turn reduces inhibition in a vicious cycle (Fig 2A3).

This explains why prolonged but not acute stress can cause MDD in susceptible individuals, and why MDD persists even after the triggering stressor is removed. Animations of these dynamics are provided in a link in Methods.

## Stress must be reduced below baseline to overcome depression due to a hysteresis effect

According to the model, MDD can be relieved by sufficiently reducing stress input. If one reduces stress input *u*, the *A* nullcline shifts down and to the left, and the *h* nullcline shifts to the right. This can abolish the MDD fixed point, leaving the euthymic state as the only possibility (Fig 2B1).

As before, this shift to low stress must last long enough (weeks) for the dynamics to cross the separatrix and converge to the euthymic fixed point. Otherwise the system returns to the MDD fixed point once stress is heightened (Fig 2B2 and 2B3).

Notably, the stress input must be reduced below the previous normal level in order to restore the healthy fixed point. This effect is known as *hysteresis*, and is caused because the positive feedback in the MDD state requires a larger than normal reduction of stress to compensate [51].

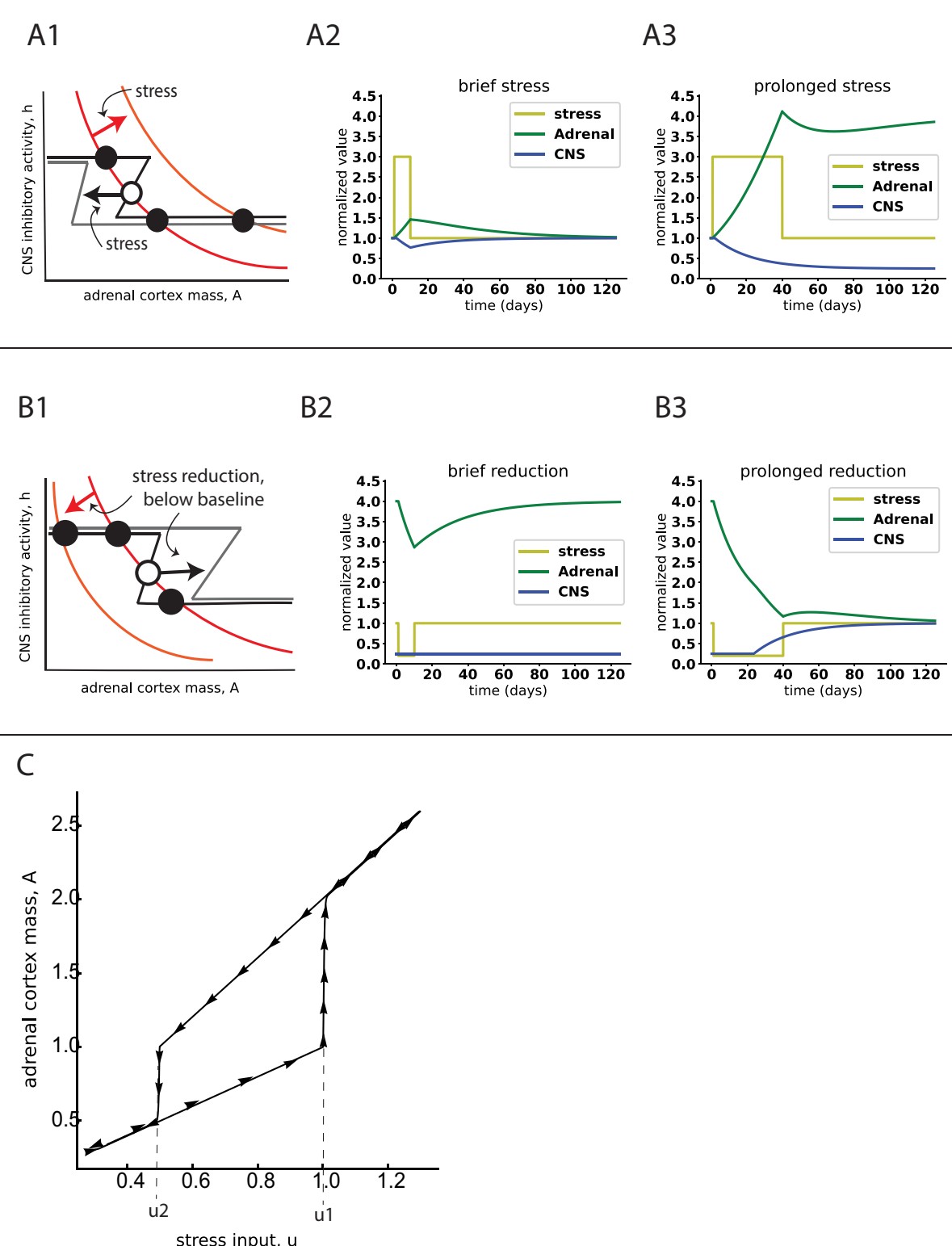

**Fig 2. Dynamics of the model show how prolonged but not acute stress can lead to depression.** A1) Stress shifts the nullclines, destroying the euthymic fixed point. A2) Dynamics under brief stress show recovery. A3) Dynamics under prolonged stress reach the depressed state on the timescale of weeks and remain there even after the stress is removed. B1) exiting depression requires stress to be reduced below baseline due to a hysteresis effect. B2) brief relief is not enough to exit depression B3) recovery on the timescale of weeks upon a prolonged stress relief. C) bifurcation plot of adrenal steady state size versus stress input $u$ shows hysteresis. The transition from euthymic state to depressed state

happens for prolonged stress levels of $u = u_1$. To return from the depressed state to the euthymic state, one must decrease stress below $u = u_2 < u_1$.

This hysteresis can be seen in a bifurcation plot, showing the steady state of $A$ versus stress $u$. Starting in the healthy state, stress must cross a threshold $u_1$ to enter the MDD state of high $A$. However, starting in the MDD state, stress must drop below a lower threshold $u_2 < u_1$ in order to return to the healthy state of low $A$ (Fig 2D).

Hysteresis may explain why in about 30% of Cushing patients with depression, depression does not abate after the tumor is treated and cortisol level decreases [11–13]. We predict these to be patients with pre-existing susceptibility to MDD, namely nullclines that cross at two stable fixed points. Cushing syndrome causes the transition to the depressed state, which remains occupied even after treatment.

## Model explains dysregulation of endocrine tests in MDD

The dysregulation of the HPA axis in MDD is characterized by two main endocrine tests, the CRH and DEX-CRH tests. One explanation for the dysregulation involves impaired GR function [29]. Here we offer a different explanation, in which GR is normal and the dysregulation is caused by enlarged glands in MDD.

The CRH test involves injecting CRH and measuring the ACTH response. Patients with MDD often show blunted ACTH responses in the CRH test relative to controls [19].

The present model explains this effect. Cortisol inhibits hypothalamic activity through MR and GR receptors, whereas it inhibits the pituitary activity through GR receptors only [49,55]. As a result, the model predicts that at the MDD fixed point, both the pituitary and the adrenal are larger than their euthymic size, and that the adrenal cortex grows more than the pituitary at steady state.

To model the CRH test, we add external CRH to the model in a pulse that lasts 30 min, corresponding to the lifetime of synthetic CRH (normal and synthetic CRH half-lives are about 4 min and 43 min respectively [45])

Control individuals are modeled with baseline stress levels and normal steady state adrenal size $A$. The CRH pulse causes an ACTH and cortisol response in both control and depressed simulations (Fig 3A1 and 3A2). However, the response is blunted in an individual with MDD, modeled with HPA glands that correspond to the MDD fixed point, namely enlarged adrenal and pituitary and high cortisol. The blunting is due to the repressive effect of high endogenous cortisol on the pituitary ACTH secretion that is not compensated by sufficient pituitary gland size.

A similar analysis explains the elevated ACTH in DEX-CRH test in depressed patients [20,21]. In this test, the long-lived cortisol analogue dexamethasone (DEX) is given at night. DEX suppresses the HPA axis. The next day, ACTH is measured after a CRH injection. Individuals with MDD show an elevated ACTH response. The model explains this due to their enlarged pituitary corticotroph mass P. The presence of DEX masks the higher cortisol found in MDD patients, and thus ACTH measured in the test corresponds primarily to the pituitary secretion capacity. MDD individuals in the model have larger P, which causes more ACTH to be secreted per CRH unit than controls (Fig 3B1, 3B2).

We note that these explanations of the tests do not require assuming that the GR receptor pathway is different in people with MDD and in controls.

## Model explains timescale of weeks for antidepressant action

We now consider the effects of changes in the $h$ nullcline. These can presumably be caused by drugs or behavioral interventions that affect the CNS. A sufficient rise in $h$ should abrogate the

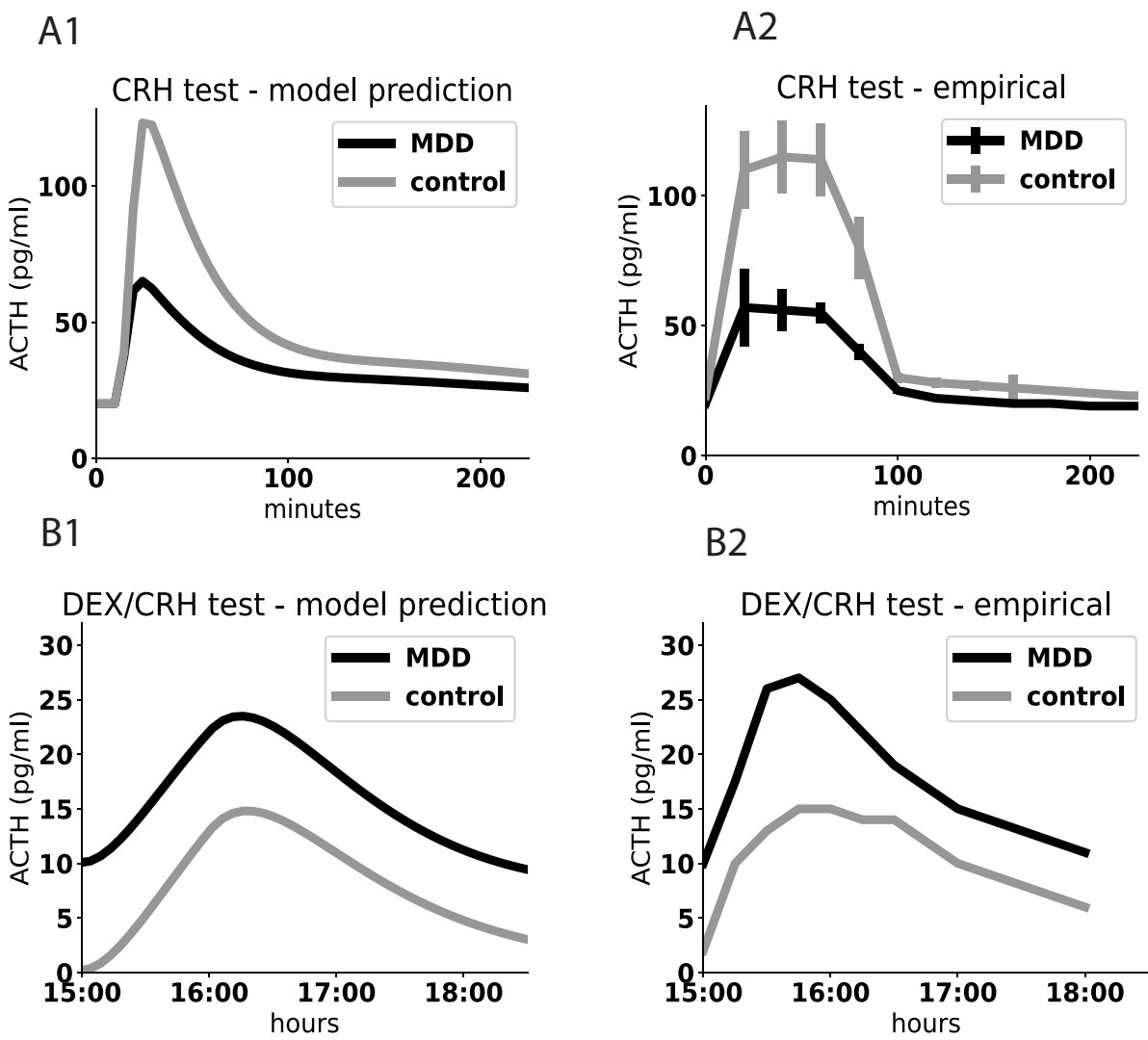

**Fig 3. Model explains CRH and DEX/CRH test aberrations in MDD based on enlarged gland sizes.** A1) Simulation shows blunted ACTH levels in CRH test in depressed patients compared to controls. A2) MDD patients show blunted ACTH in the CRH test (100μg CRH) compared to controls, adapted from Bardeleben at el [19]. B1) Simulation shows elevated ACTH levels in the DEX/CRH test in depressed patients compared to controls. B2) MDD patients show elevated ACTH in the DEX/CRH test (1.5 mg DEX and 100μg CRH 15 hours later), adapted from Heuser at el [21].

MDD fixed point (Fig 4A1) in susceptible individuals. Now the healthy point is the only possible steady state.

The slow timescale of the HPA glands, of weeks, indicates that the intervention must last at least a few weeks, otherwise the separatrix will not be crossed, and the dynamics will return to the MDD point (Fig 4A1-4A3).

This may explain why SSRIs and other medications for MDD take a few weeks to show effects [56,57]. Interestingly, SSRIs also activate the HPA axis on short timescales [58]. The model suggests they can be effective in the long term because they boost hippocampal health on the slow timescale of weeks [59]; such effects might cause the shift in the *h* nullcline that produces a therapeutic effect.

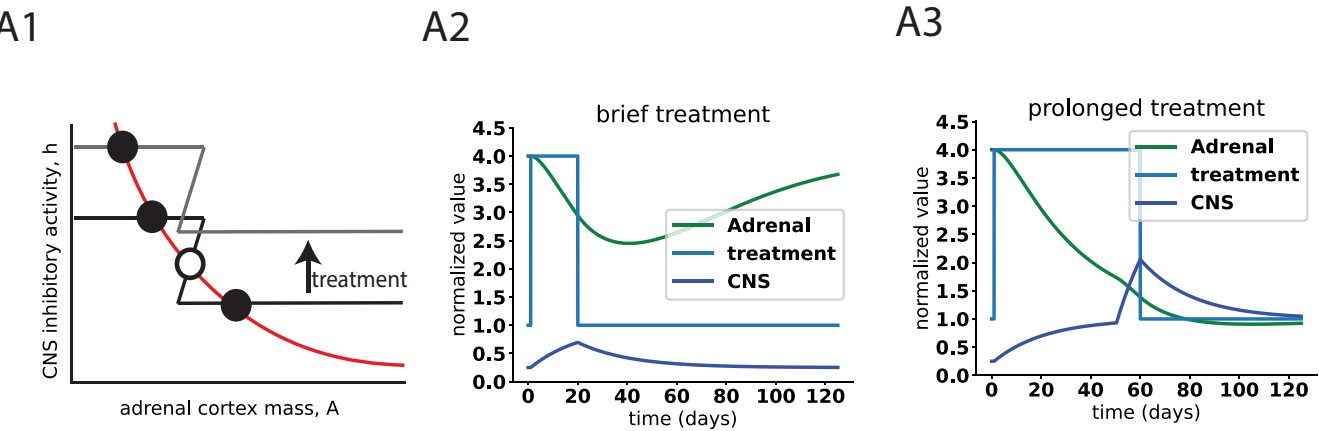

**Fig 4. Dynamics of the model may explain the timescale of weeks for antidepressant action.** A1) treatments that enhance CNS inhibitory activity can abolish the depressed fixed point. A2) brief treatment is not effective. A3) prolonged treatment of weeks or more is effective even after treatment ends.

## Discussion

We present a simple mathematical model for depression based on a CNS-HPA toggle switch which controls euthymic and depressed states on the timescale of weeks. This slow timescale is due to the turnover rate of the HPA glands. The timescale of the model explains why acute stress usually doesn't cause depression whereas prolonged stress can. It also explains why drug treatment takes weeks to affect patients. The model shows bistability, in which patients enter a self-sustaining depressed state that lasts even after stress is removed. This is due to hysteresis in which entry into the depressed state is easier than exit from this state. In addition, the model provides a mechanism for blunted CRH and elevated DEX-CRH tests in depression. This mechanism is based on HPA gland growth and differs from standard explanations based on GR receptor dysregulation.

One class of MDD drugs that seems to work without the delay of weeks is ketamine-based treatments, which often show effects within a day [60,61]. If one attempts to understand such drugs within the framework of the model, one may assume that they work in the CNS downstream of cortisol. They mitigate the negative effect of cortisol on behavior, cognition and mood, as indicated in experiments on mice models [61]. Another (nonexclusive) possibility is that ketamine affects the HPA axis directly, lowering cortisol for a given stressor, as found in male mice [62], and in a human case study in which DEX suppression tracked ketamine in a treatment-resistant patient within days [63].

The present model supplies a mechanistic explanation for the dysregulation of the HPA axis observed in MDD—namely blunted CRH and elevated DEX-CRH tests. This explanation differs from the standard explanation in which the GR receptor feedback is weaker in patients with MDD [29]. In the CRH test, ACTH is blunted because the high cortisol in MDD inhibits ACTH production in the pituitary. A subtle point is that this effect depends on the known distribution of cortisol receptors: GR in the pituitary and GR and MR in the hypothalamus. This causes the pituitary to grow in MDD but to a lesser extent than the growth of the adrenal. If both glands grew by the same factor, the CRH test would not be blunted because the large pituitary mass would compensate.

One upshot of this blunted response is that beta-endorphin secretion from the pituitary, which is thought to be proportional to ACTH secretion since both are produced from the same polypeptide POMC [64], is decreased in MDD. Beta endorphin is a euphoric and analgesic hormone. Its blunting may thus contribute to anhedonia, because a given HPA stimulus—a

given CRH pulse—causes lower levels of beta endorphin. This blunted response may contribute to the dysphoria and pain sensitivity in MDD [65,66].

In the DEX-CRH test, the extrinsic DEX acts as a cortisol analogue and masks the high endogenous cortisol level in MDD. The test exposes the functional mass of the pituitary, which is enlarged in MDD in our model. As a result, cortisol is elevated in this test in MDD relative to controls.

The model also presents a picture of susceptibility to MDD. Individuals vary in terms of their molecular/physiological parameters. This is described by the nullclines of the model. Individuals in whom the nullclines intersect three times have the potential to be stuck in MDD. Individuals for whom the nullclines intersect only once don't have the potential for MDD, and instead show a return to euthymic state even after prolonged stress. The model may, in principle, offer a way to determine susceptibility, but this would require experimental approaches to quantify the interaction strengths of the CNS and the HPA. One approach may be quantifying emotional reactivity in terms of HPA dynamics [67,68].

In summary, the HPA-CNS toggle switch model offers an understanding of aspects of MDD and their dynamics on the scale of weeks. The model is based on qualitative physiological facts; future research on the specific mechanisms can help make it more precise. The present study rationalizes susceptibility to MDD in terms of bistability of the HPA-CNS feedback loop, explains why prolonged but not acute stress can trigger MDD, why many interventions take weeks to work and how HPA dysregulation occurs in MDD. This understanding may provide concepts to guide the development of treatment strategies for MDD.

## Material and methods

### HPA-CNS toggle switch model

We model the HPA axis using the model of Karin et al [45]. We denote the input signal to the HPA system as $u$ [69]. This input is inhibited by the CNS activity $h$ [25–27], a novel feature of the present model. The concentrations of the three hormones CRH, ACTH, and cortisol are denoted $x_1$, $x_2$, $x_3$. Their fast time scale dynamics are described by (1–3):

$$\frac{dx_1}{dt} = a_1 \frac{u}{hMR(x_3)GR(x_3)} - b_1 x_1 \tag{1}$$

$$\frac{dx_2}{dt} = a_2 \frac{x_1}{GR(x_3)} P - b_2 x_2 \tag{2}$$

$$\frac{dx_3}{dt} = a_3 x_2 A - b_3 x_3 \tag{3}$$

Where P is the total pituitary corticotroph functional mass and A is the functional mass of the cortisol secreting cells in the adrenal cortex. Cortisol inhibits hypothalamic activity through MR and GR receptors, whereas it inhibits the pituitary activity through GR receptors only [33,38]. We assume that the effects of MR and GR are multiplicative and use the same MR and GR expressions as done in Karin et al [45]. MR inhibitory activity is non cooperative so that $MR(x_3) = 1 + \frac{x_3}{K_{MR}}$, whereas GR inhibitory activity is Hill like with cooperativity coefficient 3 [45,70], so that $GR(x_3) = 1 + \left(\frac{x_3}{K_{GR}}\right)^3$. We note that at physiological cortisol levels, MR receptors tend to be nearly saturated whereas GR receptors are hardly activated [39,71], namely $K_{MR} << x_3 << K_{GR}$. In this case we can approximate the MR and GR expressions—$MR(x_3) \approx x_3$,

$GR(x_3) \approx 1$. This provides a simpler set of equations for hormone levels (4–6):

$$\frac{dx_1}{dt} = a_1 \frac{u}{hx_3} - b_1 x_1 \qquad (4)$$

$$\frac{dx_2}{dt} = a_2 x_1 P - b_2 x_2 \qquad (5)$$

$$\frac{dx_3}{dt} = a_3 x_2 A - b_3 x_3 \qquad (6)$$

Next we add the equations for the functional mass of the glands. The growth factor of $P$ is $x_1$ [72,73] and the growth factor of $A$ is $x_2$ [74,75], as detailed in Karin et al [45]. Their dynamic is on a slow time scale of weeks, as a result of the pituitary and adrenal cortex turnover rates (7–8), as given by Karin et al [45]

$$\frac{dP}{dt} = a_P P x_1 - b_P P \qquad (7)$$

$$\frac{dA}{dt} = a_A A x_2 - b_A A \qquad (8)$$

The main new feature of the present model is the variable $h$ that represents CNS inhibitory activity. We posit a slow time scale of weeks for $h$ in Eq (9) by setting its removal rate appropriately. We note that this choice does not affect the conclusions: a timescale of hours or days for h would leave the qualitative results unchanged, and only have mild effects on the dynamics. The relevant CNS regions, hippocampus and frontal cortex, express both high-affinity MR and low-affinity GR receptors for cortisol. The GR generally has inhibitory effects. As mentioned above, MR receptors are nearly saturated by cortisol in its physiological range [71]. The GR receptors have a cooperative response to cortisol which is approximately step-like. We therefore, for simplicity, model the inhibitory effect of cortisol on the CNS as a step function, defined by $a+b\Theta(x_3>T)$. Using a Hill-type function instead of a step function leaves the conclusions unchanged but makes the analytical solution much more difficult. When cortisol level is lower than the threshold $T$, only MR receptors are activated, causing CNS inhibition by a factor of $a$. But when cortisol level exceeds the threshold $T$, both MR and GR are activated, inhibiting CNS by a factor of $a+b$.

$$\frac{dh}{dt} = a_h \frac{1}{a + b\Theta(x_3 > T)} - b_h h \qquad (9)$$

Our HPA-CNS model is the set of equations (Eqs 1,2,3,7,8,9) which describe the dynamics of hormones and glands in the HPA-CNS system.

In order to demonstrate the general features of the model and nullclines, we first use a simplified version (Eqs 4,5,6,7,8,9). When considering compensation mechanisms where the GR plays an important role, we use the full HPA-CNS model Eq (Eqs 1,2,3,7,8,9).

The nominal parameter values for the model are presented in Table 1. These parameters provide 3 fixed points, and provide hormone levels equal to 1. The fold-change of each parameter required to bifurcate to a single fixed point are also shown. Some parameters cause a bifurcation both when raised and lowered, as represented by two fold-change values in the table.

The equations have two timescales—Eqs (4–6) have a timescale of hours, whereas Eqs (7–9) have a timescale of weeks. Solving the fast equations at quasi-steady-state (qss) gives the fast

time levels of the hormones $x_1$, $x_2$, $x_3$ (10–12):

$$x_{1_{qss}} = \left(\frac{a_1 b_2 b_3}{b_1 a_2 a_3} \frac{u}{hAP}\right)^{\frac{1}{2}} \propto \left(\frac{u}{hAP}\right)^{\frac{1}{2}} \tag{10}$$

$$x_{2_{qss}} = \left(\frac{a_1 a_2 b_3}{b_1 b_2 a_3} \frac{Pu}{Ah}\right)^{\frac{1}{2}} \propto \left(\frac{Pu}{Ah}\right)^{\frac{1}{2}} \tag{11}$$

$$x_{3_{qss}} = \left(\frac{a_1 a_2 a_3}{b_1 b_2 b_3} \frac{APu}{h}\right)^{\frac{1}{2}} \propto \left(\frac{APu}{h}\right)^{\frac{1}{2}} \tag{12}$$

where one assumes that gland sizes are constant on the hour timescale.

Solving Eqs 7 and 8 at steady state shows that the slow time steady state of $x_1$ and $x_2$ is quite simple, and independent on the input u and most other model parameters, due to the properties of integral feedback in these equations [45].

$$x_{1_{st}} = \frac{b_P}{a_P} \tag{13}$$

$$x_{2_{st}} = \frac{b_A}{a_A} \tag{14}$$

We normalize our equations $x_1 := \frac{x_1}{x_{1_{st}}}$, $x_2 := \frac{x_2}{x_{2_{st}}}$, so the baseline level is now $x_{1_{st}} = x_{2_{st}} = 1$. Eqs 13 and 14 show that upon a step change in input u, the change in hormone levels $x_1$ and $x_2$ is transient and the pituitary and adrenal glands grow to compensate to return CRH and ACTH to their original baseline levels within weeks. This growth and compensation can be calculated by substituting $x_{1_{st}}$, $x_{2_{st}}$ in Eqs (5–6). From Eq (5) we find that the adrenal gland steady-state sizes rises linearly with steady-state cortisol $A_{st} \propto x_3$. On the other hand, from Eq (6) we find that the pituitary's steady state size is independent on cortisol level, $P_{st} = \frac{b_A a_P b_2}{a_A b_P a_2}$. Below we will show that if we solve the original Eqs (1–3) and consider GR inhibitive effect on the pituitary, we find a compensation mechanism from both adrenal and pituitary glands.

Now, if we substitute $x_{1_{st}}$ in Eq (4), we find that the steady state of cortisol is

$$x_{3_{st}} = \frac{a_1 a_P}{b_1 b_P} \frac{u}{h} \propto \frac{u}{h}.$$

This makes biological sense by having cortisol proportional to its hypothalamic input signal u. To obtain the HPA-CNS model slow time steady state solution, we substitute fast time scale solutions (10–12) into the slow Eqs (7–9):

$$\frac{dP}{dt} = a_P P \left(\frac{u}{hAP}\right)^{\frac{1}{2}} - b_P P \tag{15}$$

$$\frac{dA}{dt} = a_A A \left(\frac{Pu}{Ah}\right)^{\frac{1}{2}} - b_A A \tag{16}$$

$$\frac{dh}{dt} = a_h \frac{1}{a + b\Theta\left(\left(\frac{APu}{h}\right)^{\frac{1}{2}} > T\right)} - b_h h \tag{17}$$

In order to demonstrate this slow steady state solution space in a two-dimensional phase plane, we use the pituitary's steady state size, $P_{st} = \frac{b_A a_P b_2}{a_A b_P a_2}$, removing one of the three equations

and allowing a 2D analysis. This allows us to solve $\frac{dh}{dt} = 0$ and $\frac{dA}{dt} = 0$ to find the slow time scale steady state of the system, and to understand the shape of $h$ and $A$ nullclines (18–19):

$$\frac{dA}{dt} = 0 \rightarrow h = \left(\frac{a_A}{b_A}\right)^{\frac{1}{2}} P_{st} \frac{u}{A} = \left(\frac{a_A}{b_A}\right)^{\frac{1}{2}} \frac{b_A a_P b_2}{a_A b_P a_2} \frac{u}{A} \propto \frac{u}{A} \tag{18}$$

$$\frac{dh}{dt} = 0 \rightarrow h = \frac{a_h}{b_h} \frac{1}{a + b\Theta(x_3 > T)} = \frac{a_h}{b_h} \frac{1}{a + b\Theta\left(\left(\frac{AP_{st}u}{h}\right)^{\frac{1}{2}} > T\right)} \tag{19}$$

These two nullclines can generate a bistable system as seen in Fig 1.

Now, we explore the compensation mechanisms of the system, using the steady state solution of (2–3) and (7–8). This compensation can be estimated by substituting $x_{1_{st}}, x_{2_{st}}$ in Eqs (2–3) steady state solutions. We find that $A_{st} \propto x_3$ whereas $P_{st} \propto GR(x_3) = 1 + \left(\frac{x_3}{K_{GR}}\right)^3$. Both the pituitary and adrenal cortex grow in order to compensate for high CRH and ACTH levels, but the adrenal cortex grows more than the pituitary.

Interventions and drugs that affect the CNS are modeled by increasing CNS inhibitory activity by a factor of D:

$$\frac{dh}{dt} = a_h \frac{D}{a + b\Theta(x_3 > T)} - b_h h \tag{20}$$

### Animation of phase portrait under parameter changes

Animations, generated by Mathematica, of the phase portraits under changing u, a or b parameters can be found here: https://github.com/benron20/MDD-and-bistability-in-the-HPA-stress-axis.

## Supporting information

The Supporting Information, including all code files and animations can be found here: https://github.com/benron20/MDD-and-bistability-in-the-HPA-stress-axis.

**S1 File. Hysteresis.**
(PY)

**S2 File. CRH test graph.**
(PY)

**S3 File. DEX/CRH test graph.**
(PY)

**S4 File. CRH test data.**
(PY)

**S5 File. DEX/CRH test data.**
(PY)

**S6 File. HPA phase portrait.**
(IPYNB)

**S7 File. Dynamic of brief stress.**
(PY)

**S8 File. Dynamic of prolonged stress.**
(PY)

**S9 File. Dynamic of brief stress reduction.**
(PY)

**S10 File. Dynamic of prolonged stress reduction.**
(PY)

**S11 File. Dynamic of short treatment.**
(PY)

**S12 File. Dynamic of prolonged treatment.**
(PY)

## Author Contributions

**Conceptualization:** Ben Ron Mizrachi, Avichai Tendler, Omer Karin, Tomer Milo, Uri Alon.

**Data curation:** Ben Ron Mizrachi, Dafna Haran, Uri Alon.

**Formal analysis:** Ben Ron Mizrachi, Avichai Tendler, Omer Karin, Tomer Milo, Avi Mayo, Uri Alon.

**Funding acquisition:** Uri Alon.

**Investigation:** Avichai Tendler, Dafna Haran, Avi Mayo.

**Methodology:** Ben Ron Mizrachi, Avichai Tendler, Omer Karin, Avi Mayo, Uri Alon.

**Project administration:** Uri Alon.

**Resources:** Uri Alon.

**Software:** Ben Ron Mizrachi, Avichai Tendler, Avi Mayo.

**Supervision:** Uri Alon.

**Validation:** Tomer Milo, Uri Alon.

**Visualization:** Ben Ron Mizrachi, Avi Mayo.

**Writing – original draft:** Ben Ron Mizrachi, Uri Alon.

**Writing – review & editing:** Ben Ron Mizrachi, Uri Alon.

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
