## [Decision Letter · Decision Letter 0]

19 Jul 2023

Dear prof Alon,

Thank you very much for submitting your manuscript "Major depressive disorder and bistability in an HPA-CNS toggle switch" for consideration at PLOS Computational Biology. As with all papers reviewed by the journal, your manuscript was reviewed by members of the editorial board and by several independent reviewers. The reviewers appreciated the attention to an important topic. Based on the reviews, we are likely to accept this manuscript for publication, providing that you modify the manuscript according to the review recommendations.

Sincerely,

James R. Faeder

Academic Editor

PLOS Computational Biology

Pedro Mendes

Section Editor

PLOS Computational Biology

Reviewer's Responses to Questions

**Comments to the Authors:**

Reviewer #1: "Major depressive disorder and bistability in an HPA-CNS toggle switch" (PCOMPBIOL-D-23-00795) by prof Uri Alon and colleagues submitted to PLOS Computational Biology is a well written and important paper continuing earlier contributions by the same authors.

Overall, the core part does only require minor changes whereas the Method section and especially the presentation of the mathematical model and the analysis of this needs extensive work.

P.3 Results has a subsection named ‘A minimal model of the HPA-CNS toggle switch’ but in the Method section this is denoted the MDD model. You need to decide upon a single name! Despite your choice an appropriate and more descriptive heading could be ‘A toggle switch model of the HPA-CNS system’ (In the text the ‘minimal model’ feature is mentioned).

P.3, l.-5 (meaning line 5 from bottom of page 3). I suggest substituting the word ‘described’ by ‘understood’.

P.4 and throughout the paper, the imprecise and mathematically misleading term ‘steady state’ is used for ‘quasi-steady state’ (which could be abbreviated QSS). Please correct this throughout the text.

P.4 , l.-6 ‘Kd’ is not defined. You may explain it in words.

P.4, l.-4 It is not clear to the readers what ‘our model’ is!!!?

P.5, Figure 1. Remove the vector field or make it more visible and complete (it appears sparce on the phase plane).

P.6 Figure 1 caption. It is not clear how C) and D) actually are made, state if C) and D) correspond to high and low u.

P.8, Figure 2. Strange idea to change colors compared to figure 1 (this is also the case for figure 4). Please unify.

P.12, l.10 I suggest interchanging the word ‘minimal’ with ‘simple’.

P. 14, l.2 You refer to nicely to Karin et al. However, the equations (1-3) looked very much like some I saw year before this reference. Thus, I looked it up and to me it seems like Karin et al is inspired by Bangsgaard & Ottesen (2017) Patient specific modeling of the HPA axis related to clinical diagnosis of depression. Math. Biosciences, 287. The authors also have a reduction like the one your do leading to (4-6). I think it would be suitable to refer to this paper.

P.14, l.3 Put in bio-refs to support the claim or state that it is an assumption.

P.14, eq. (1) Can you provide the reader with a bio-reference saying that MR and GR act multiplicatively? If not emphasize that this is an assumption.

P.14, l.12-13. Are the expressions for MR(x_3) and GR(x_3) assumed or are these coming from data of some kinds?

P.14, l.16. I think you need to be specific about assuming GR=1 and MR=x_3 to arrive with the reduced system (4-6). Likewise, you should emphasize that these are only good approximations for K_MR<<x_3<<k_gr>P. 14, l.21-22. Why the choice x_1 as growth factor for P and x2 for A? I would imagine that the hormone produced generates the growth, meaning that it should be x_ and x_3 respectively (this is common in many similar models). Give a bio-reference for this or emphasize these a (strange) assumptions!

P. 14, l.-3 ‘to a good approximation’ compared to what? Maybe just write ‘approximately’?

P.14, l.-1-P.15, l.2. I think this is pseudo modeling, since you are really modeling the common MR and GR inhibition as a step function?

P.14-P.16 You may use the terms fast QSS and slow QSS consistently throughout this section!

P.15, eq (9) First term on the right-hand side is really alpha-beta*theta(x_3>T) for suitable choices of positive beta<alpha. complication="" reciprocal="" so="" the="" why="">P.15, l.12 Is there an ‘of’ too much?

P.15, l.17 ‘normalize’ do you mean \\tilde{x}_1=x_1/x_st, etc. and then renaming \\tilde{x}_1 as x_1 etc.?

P.15, l.17-23. Is sloppy hand-waving / non-rigid and intuitive. Either that should be crystal clear or done properly e.g., does a QSS approximation require a stability analysis!

P.16, l.3-6. This is not rigid but merely an illustration.

P.16, eq. (18) Should the power 1/2 be the power 2?

p.16, l. 12 ‘We’ after the period.

P.16, l.13 Pst proportional to GR(x_3), but earlier you did assume GR(x_3)=1!

P.16. After reading the section I still do not know what your model exactly is. Please state the resulting model uniquely!

P.16. Add a table with all parameter values used.</alpha.></x_3<<k_gr>

Reviewer #2: Review of Mizrachi et al, Major depressive disorder and bistability in an HPA-CNS toggle switch

This paper addresses the origin of slow (weeks long timescale) dynamics of major depressive disorder using a minimal mathematical model and dynamical systems approaches. The model focuses on the interaction between the hypothalamus-pituitary-adrenal axis (HPA) and the central nervous system (CNS), and specifically incorporates the important but overlooked phenomena by which activation can control hormone gland sizes. The authors focus on the role of mutual inhibition between CNS and HPA activity, the latter mediated by glucocorticoids.

Overall this work beautifully exemplifies ability of carefully crafted simplified dynamical models to provide powerful insights into the behavior of complex biological systems. It suggests an intriguing mechanism that explains how different setpoints in individuals can produce distinct (bistable vs. monostable) dynamics, how hysteretic effects can lead to extended timescales, and how specific disease-relevant perturbations could impact system dynamics. I anticipate it will have a major impact in the field of systems psychiatry, and will be of broader interest to researchers in systems biology, systems medicine, and psychiatry more generally. Given the increasing awareness and incidence of depression, especially among teens, I believe this work is also timely and potentially of interest beyond the academic and medical communities. The paper is beautifully and clearly written. I support acceptance in its current form. However, below I suggest a few minor comments the authors may wish to consider.

Minor comments:

• Cosmetic: Some of the figures have oddly compressed fonts, or somewhat fuzzy (low resolution) graphics, possibly reflecting an issue with the image compression.

• It would be helpful to understand which parameters, if any, have the strongest impact on the qualitative predictions.

• The authors may wish to consider making an animation of the nullcline analysis in which the reader could directly see how the nullclines and the system state “move” during various perturbations discussed in the text and, e.g. in Figure 2.

Reviewer #3: This is a very important study that can help inform the treatment modalities for major depressive disorders, which afflict millions of people worldwide. By focusing on the HPA axis, which has a well documented role in MDD, the authors develop a mathematical model that describes the consequence of the HPA axis design (with two negative feedbacks and the resulting toggle switch) and importantly explain the requirement for long time scale of treatment to produce therapeutic effects. Although the study intentionally focused on one aspect of a complex problem, it provides valuable insights into the dynamics of HPA axis and explain previously poorly understood characteristics of MDD (requirement for prolonged stress) and its treatment (requirement for prolonged use of certain classes of anti-depressants).

I think overall this is a very valuable contribution with original insights that would benefit the community.

**Have the authors made all data and (if applicable) computational code underlying the findings in their manuscript fully available?**

Reviewer #1: **No: **I could find any aailable code

Reviewer #2: Yes

Reviewer #3: Yes

PLOS authors have the option to publish the peer review history of their article (what does this mean?). If published, this will include your full peer review and any attached files.

Reviewer #1: No

Reviewer #2: No

Reviewer #3: No

Figure Files:

Data Requirements:

Reproducibility:

References:

---

## [Decision Letter · Decision Letter 1]

1 Nov 2023

Dear prof Alon,

We are pleased to inform you that your manuscript 'Major depressive disorder and bistability in an HPA-CNS toggle switch' has been provisionally accepted for publication in PLOS Computational Biology.

Before your manuscript can be formally accepted you will need to complete some formatting changes, which you will receive in a follow up email. A member of our team will be in touch with a set of requests. Please also note that Reviewer 1 has noted a few instances of formatting errors or missing elements that need to be corrected.

Best regards,

James R. Faeder

Academic Editor

PLOS Computational Biology

Pedro Mendes

Section Editor

PLOS Computational Biology

Reviewer's Responses to Questions

**Comments to the Authors:**

Reviewer #1: The manus is a litle sloppy and need some care in the next phase. This regard many references without authors (internet links appear) and some phrases in the text e.g. "parameters xxx.”" at page 24

**Have the authors made all data and (if applicable) computational code underlying the findings in their manuscript fully available?**

Reviewer #1: Yes

PLOS authors have the option to publish the peer review history of their article (what does this mean?). If published, this will include your full peer review and any attached files.

Reviewer #1: **Yes: **Johnny T. Ottesen

---

## [Editor Report · Acceptance letter]

13 Nov 2023

PCOMPBIOL-D-23-00795R1 

Major depressive disorder and bistability in an HPA-CNS toggle switch

Dear Dr Alon,

I am pleased to inform you that your manuscript has been formally accepted for publication in PLOS Computational Biology. Your manuscript is now with our production department and you will be notified of the publication date in due course.

With kind regards,

Judit Kozma
